# Control of Droplet Transition in Underwater Welding Using Pulsating Wire Feeding

**DOI:** 10.3390/ma12101715

**Published:** 2019-05-27

**Authors:** Ning Guo, Lu Huang, Yongpeng Du, Qi Cheng, Yunlong Fu, Jicai Feng

**Affiliations:** 1State Key Laboratory of Advanced Welding and Joining, Harbin Institute of Technology, Harbin 150000, China; hlu621@126.com (L.H.); duypgg@163.com (Y.D.); 13933851375@163.com (Q.C.); 15554470788@126.com (Y.F.); fengjc@hit.edu.cn (J.F.); 2Shandong Provincial Key Laboratory of Special Welding Technology, Harbin Institute of Technology at Weihai, No. 2 Wenhuaxi Road, Weihai 264209, China; 3Institute of Oceanographic Instrumentation, Shandong Academy of Sciences, Qingdao 266001, China

**Keywords:** pulsating wire feeding, frequency, duty cycle, withdrawal

## Abstract

Underwater wet welding technology is widely used. Because the stability of droplet transfer in underwater wet welding is poor, the feasibility of improving the droplet transfer mode has been discussed from various technical directions. In this work, the characteristics of pulsating wire feeding were studied in the pulsating wire feeding mode by investigating the effects of changing the pulsating frequency, the wire withdrawal speed, and the wire withdrawal quantity on the droplet transfer process and the welding quality. With the aim of improving weld forming and welding stability, the authors selected the coefficient of variation and the ratio of unstable droplet transfer as the indexes to evaluate the effect of droplet transfer control. The pulsating wire feeding process of underwater wet flux-cored wire was analyzed in depth, and the following conclusions were drawn: using the pulsating wire feeding mode and after comparing and analyzing the pulsed wire feeding process under the same frequency condition, the authors found that the forming and stability were better under the conditions of slower withdrawal speed and smaller withdrawal quantity. The short-circuit transition ratio decreased steadily with the increase of pulsating wire feeding frequency, the rejection transition ratio first rose and then decreased, and the splash ratio first decreased and then rose.

## 1. Introduction

With the fast-developing exploitation of marine resources, underwater welding technology is becoming more and more important [1,2]. Underwater welding technologies based on traditional fusion welding methods in air have developed greatly in recent years, but trials and engineering applications demonstrate that troublesome issues still remain with underwater wet fusion welding, especially in the deep sea, such as hydrogen embrittlement, arc instability, etc. [3,4,5,6]. Underwater wet arc welding technologies mainly include underwater wet covered electrode arc welding (shielded metal arc welding, or SMAW) and underwater wet self-shielded flux-cored arc welding (FCAW-S). Among them, the underwater wet FCAW-S welding technology has attracted more and more attention because of its high welding efficiency and its applicability to automatic welding [7]. Sun et al. developed ultrasonic-assisted underwater wet welding equipment, including a welding system, an ultrasonic system, and a composite torch. By forming the ultrasonic radiation force on the bubbles during real-time welding, the ultrasonic energy can be effectively applied to improve the arc stability and enhance the mechanical properties of the joints [8]. Kong et al. introduced in detail a method for measuring the fluid discharge of diffusible hydrogen in weldments. They found that the diffusible hydrogen content of the underwater welding electrode was higher than that of atmospheric welding by using the fluid discharge method. The results are closely related to welding materials. Therefore, the best way to control diffusible hydrogen is to adjust the welding materials and improve slag fluidity [9]. Świerczyńska et al. ascertained the diffusible hydrogen content in deposited metal using the glycerin method according to the Plackett–Burman design, determining the significance of the effect of the stick-out length, welding current, arc voltage, travel speed, and water salinity. In addition, they summarized the effect of underwater wet welding parameters and conditions on the diffusible hydrogen content in deposited metal for welding with a self-shielded flux-cored wire [10]. At the same time, compared with air, the thermal conductivity of water is relatively high. The water environment will greatly increase the cooling rate of the weld zone and the cooling speed can reach 6–8 times that of land welding [11]. Currently, the problem of cold cracking in underwater wet welding can be controlled from the two aspects of welding material and process. The composition of the welding slag system is adjusted to reduce the adverse effect of the hydrogen-rich environment on wet underwater welded joints and avoid the occurrence of welding defects such as gas holes and cold cracking in welded joints [12]. In addition, adjusting the proportion of alloy components in covering and core can optimize the microstructures of welded joints, improve the toughness of welded joints, and prevent the occurrence of cold cracking. Furthermore, better methods have been applied to underwater wet welding to control the cold cracking during wet welding. For example, the temper bead welding (TBW) technique is multi-layer and multi-pass welding. The heat-affected zone (HAZ) of the former bead is tempered by the thermal cycle of the latter bead to reduce the hardening structure of the heat-affected zone and improve the mechanical properties of the welded joint [13]. Tomków et al. knew from experiments that S355G10+N steel is highly sensitive to forming cold cracks in a water environment, and TBW technology is an effective way to improve the weldability of steel. It has been verified that the tempered bead technology can effectively reduce the sensitivity of cold cracks in a water environment [14]. Tomków et al. studied the effect of the tempered bead technique on the weldability of the steel studied. Microcracks can be found in the HAZ of all welded joints by microscopic testing. Using the tempered bead technique as a method to improve the weldability of steel, the maximum hardness of HAZ can be reduced to below the critical value of 380 HV10 [15]. Affected by the heat dissipation, environmental pressure and a large amount of hydrogen, the shape of the underwater welding arc changes significantly, and the arc static characteristic curve is also quite different from the onshore welding process [16]. 

Currently, there are four methods to study droplet transfer, namely, acoustic signal analysis, spectral analysis, electrical signal analysis, and visual analysis. Among them, the visual analysis method is the most intuitive way to detect the droplet transfer, which directly observes the image of the droplet transfer [17]. Electrical signal analysis is often used to study arcing and droplet behavior during welding, which is directly related to the droplet transfer and easy to collect. Synchronous arc electrical signal in high-speed video recording is a commonly used test and analysis method in the study of droplet transfer in onshore welding process.

In order to improve the droplet transfer process of underwater wet welding, improvements in welding materials, welding technology, and welding equipment are mainly carried out. Bayley and Mantei studied the welding process of high-strength steel and found that because the heat input of the weld is relatively high and the cooling rate is slow, the conventional arc welding process can produce a softening heat-affected zone, which usually results in a lower strength of the welded joint [18,19]. Gao et al. pointed out that temper bead welding technology is one of the most widely used welding technologies without any complicated underwater equipment, and studied underwater wet metal arc welding technology using three kinds of heat input. The results showed that the weld metal has the best mechanical properties with a small heat input due to the greatest number of weld passes associated with the largest fraction of the fine reheated zone. However, at the same time, it also increases the tendency of cold cracking of the coarse grain heat-affected zone (CGHAZ) and makes the controlling of interval time between passes stricter when applying TBW. In addition, in multi-layer welding, temper bead can improve the brittle microstructure, lower the hardness, and reduce the hydrogen-induced cracking tendency [20]. Maksimov studied the continuous cooling transformation diagram of X60 steel and the phase composition and structure of the heat-affected zone of an underwater welded joint. The analysis showed that the heating and cooling conditions in underwater welding have a controlling effect on the structure and properties of the welded joint in comparison with welding in air. An increase in the cooling rate by almost an order of magnitude reduces the dimensions of the HAZ and increases hardness, and quenched structures appear in the material, that is, the degradation of the structure and the mechanical properties, controlled by the structure, takes place in the welded joint [21]. Łabanowski et al. used the gas metal arc (GMA) local dry welding method to make welded joints of duplex stainless steel, which proved that duplex stainless steel had a good underwater local dry welding property [22]. Liu et al. researched the metal transfer behavior of self-shielded flux-cored wire and found that the metal transfer mode included bridging transfer, globular repelled transfer, and droplet transfer [23]. Guo et al. observed the metal transfer process of underwater flux-cored wet welding using an X-ray transmission method, and found four fundamental metal transfer modes, namely, the globular repelled transfer mode, the surface tension transfer mode, the explosive short-circuit transfer mode, and the “submerged arc transfer mode” [24,25]. When Cranfield University studied the influence of pulsating current parameters on the welding process, it was found that when the peak current was 220–240 A, the spatter amount could be controlled, and the increase of peak current had little effect on the droplet transition process. The duration of peak current should be controlled at 3–4 ms; a duration too long or too short would significantly affect the droplet transition process, and the amount of operation spatter would increase [26]. 

For this paper, the method of wire feeding was changed from constant speed wire feeding to pulsating wire feeding. In this regard, we investigated the effect of pulsed wire feeding on the droplet transfer process of underwater welding under different pulse frequencies and pulse duty ratios, and explored the control method of droplet transfer to improve the quality of underwater wet welding. Through design experiments, the effects of pulsating frequency, wire withdrawal speed, and wire withdrawal quantity on the quality of pulsating wire feeding were studied. The influence of the pulsating wire feeding mode on the droplet transfer process of underwater welding was examined to increase the quality of underwater wet flux-cored wire welding by improving the control strategy for pulsating wire feeding.

## 2. Materials and Methods 

This subject requires that the conditions of different droplet transfer rates in the welding process be explored. Therefore, we had to observe and count the transition ratios of the droplets under different conditions. For this purpose, an underwater welding X-ray high-speed camera system (Dandong Zhongxun Technology Co., Ltd, Dandong, China) was introduced. This system was composed mainly of a microfocus X-ray source, a special welding tank, an image intensifier, a high-speed camera, a computer, and a lead room, as is shown in Figure 1.

The welding surfacing test specimens (Anshan Iron and Steel Group, Anshan, China) used in the experiment were made of Q235 mild steel and the dimensions were 200 mm × 50 mm × 15 mm. The chemical composition of Q235 steel is shown in Table 1. The welding material is a rutile-type underwater wet welded flux-cored wire with a diameter of 1.6 mm obtained from the E. O. Paton Electric Welding Institute (Kiev, Ukraine), which consists of rutile, marble, feldspar, fluorite, alumina, and lithium fluoride mixed in a specific proportion. In this experiment, the welding polarity mode chosen was direct current electrode positive (DCEP), and 32 V of arc voltage was selected to work under constant voltage.

In this experiment, the effects of pulsating wire welding on the stability of underwater wet welding and the transition state of droplets were investigated. Therefore, the weld forming score, the coefficient of variation, and the proportion of transition mode were selected as the indexes for evaluating the effect of the welding.

During underwater welding, the weld is often accompanied by slag inclusion and gas holes, which have a bad effect on the weld. Therefore, we divided the weld forming into four grades to evaluate the weld effect, each of which corresponded to a score. The highest score was 3 points, which corresponded to a complete and continuous weld without arc extinguishing, indicating that the weld was well formed. The lowest score was 0, which corresponded to many defects such as discontinuity and spatter, indicating that the weld was difficult to form.

When arc breaking or a short circuit occurs during underwater welding, the distance between the wire and workpiece changes sharply, and the arc voltage changes significantly. When spatter occurs, the pool vibrates violently and the arc voltage also changes greatly. Therefore, the arc voltage’s coefficient of variation was selected to characterize the stability of the welding process. The bigger the coefficient of variation is, the more unstable is the welding process. On the contrary, the smaller the coefficient of variation is, the more stable is the welding process.

Droplet transfer in underwater welding can be divided into three types: rejection transition, short-circuit transition, and spatter transition. Rejection transition refers to the exclusion of droplets from the welding wire when they are near the molten pool, and then they drop down into the molten pool. Short-circuit transition means that a short-circuit current is generated when the droplets come into contact with the welding wire and the molten pool, which causes them to fall into the molten pool. Finally, splash transition refers to the droplets being bombed into the molten pool area when s short-circuit explosion occurs or the rejection force is too strong.

## 3. Results

In order to further suppress the proportion of the transition process and improve the stability of the welding process, the wire feeding process can be introduced on the basis of the pulsed wire feeding technology, and the wire feeding mode of “one delivery, one stop” is converted into the “one delivery-one pumping and one stop” mode to speed up the droplet transfer process.

The change of wire feeding speed during the pulsating wire feeding process is shown in Figure 2, where *V*_1_ is the wire feeding speed, *T*_1_ is the duration of the wire feeding process in one wire feeding cycle, *V*_2_ is the wire withdrawal speed, *T*_2_ is the wire withdrawal time in one wire feeding cycle, and *T*_3_ is the time when the wire feeder stops running in one wire feeding cycle.

In the pulsating wire feeding process, the average wire feeding speed can be calculated using Equation (1) as follows:(1)$$V_{avr}=\frac{V_{1}T_{1}-V_{2}T_{2}}{T_{1}+T_{2}+T_{3}}.$$

Since the wire withdrawal time and the wire dwell time are introduced, the wire feed frequency can be calculated using Equation (2) during the pulsating wire feeding process as follows:(2)$$F=\frac{1}{T_{1}+T_{2}+T_{3}}.$$

In the pulsating wire feeding process, as long as the wire feeding stage can effectively convey the welding wire, the wire is not fed forward in the excessive phase and the stopping phase. Therefore, the duty ratio of the pulsating wire feeding process can be calculated using Equation (3) as follows:(3)$$D=\frac{T_{1}}{T_{1}+T_{2}+T_{3}}.$$

It should be pointed out that the arc will be elongated during the withdrawal process. Under the premise that the arc voltage is relatively stable, the arc is too long to be easily extinguished. Therefore, it is necessary to properly control the wire withdrawal speed and the withdrawal time to prevent arc extinguishing due to excessive withdrawal distance. Wire withdrawal distance can be calculated using Equation (4) as follows:(4)$$L_{hc}=V_{2}T_{2}.$$

In order to investigate the effects of wire feeding speed, withdrawal quantity, and pulsating frequency for the pulsating wire feeding welding process, experiments were designed as shown in Table 2.

As the effect of wire withdrawal is equivalent to that of over-heating under the condition of pulsed wire feeding, the arc extinguishing effect occurs. When the frequency of pulsating welding increases, the frequency of arc extinguishing increases accordingly. Frequent arc extinguishing has a strong impact on the stability of the welding. During underwater welding, the self-shielding gas around the welding wire itself has an obvious fluctuation effect and bubbles float periodically. When the arc is interrupted frequently, the water directly impacts the end of the welding wire and the liquid molten pool area, so the welding effect becomes worse. When welding at 10 Hz frequency, the phenomenon of reburning and arc extinguishing still occurs, but the welding forming effect is the best. At 60 Hz, the weld is extremely unstable, and it is difficult to produce continuous and stable welds. This situation is shown in Figure 3, Figure 4 and Figure 5.

The influence of the pulsating wire feeding frequency on the weld formation is shown in Figure 6. As the pulsating wire feeding frequency increases, the welding forming points have a significant downward trend, and the constant speed wire feeding group without the withdrawal is used as the 0 Hz control. The pulsating wire feeding formation was significantly reduced, and the downward trend was obvious.

The coefficient of voltage variation is shown in Figure 7. There is no obvious regularity with the increase of pulsating wire feeding frequency. When the weld forming score has a higher value at 10 Hz, the coefficient of voltage variation reaches its maximum value, which is inconsistent with the stability of the welding process under certain conditions. It is concluded that the former pulsating wire feeding mode itself may have a great influence on the arc length. The coefficient of voltage variation cannot reflect the welding stability of the pulsating wire feeding process.

Short-circuit transition decreases steadily with the increase of pulsating wire feeding frequency. As shown in Figure 8, it is in accordance with the phenomenon that the fast welding wire in the wire feeding stage of low-frequency pulsating wire feeding obtained by high-speed video cannot melt and easily penetrate into the molten pool to form short-circuit transition. With the increase of pulsating wire feeding frequency, the rejection transition ratio first rises and then decreases, and the splash ratio first decreases and then rises. Because the molten pool splashes more at 60 Hz, the splash ratio is increased. The droplet splashes out of the molten pool area, caused by the violent oscillation of the molten pool, reach the largest proportion at 60 Hz; the splash ratio is less at 40 Hz, and the main mode of droplet transition is the rejection transition.

The 50% duty cycle pulsed wire feeding group without withdrawal was taken as the control group with 0 withdrawal speed. Figure 9 shows that the forming score of pulsating wire feeding decreases with the increase of withdrawal speed. Under 500 m/h withdrawal speed, the weld forming score is better than that without withdrawal with the same duty cycle pulsed wire feeding. At 1000 m/h withdrawal speed, the weld forming score is the lowest. Figure 10 indicates that the coefficient of variation increases with the increase of wire withdrawal speed. The coefficient of variation in the pulsating wire feeding experimental group is significantly higher than that in the pulsed wire feeding control group. It shows that the pulsed wire feeding back has a great influence on arc length, and the coefficient of variation cannot be used as an index to evaluate the welding stability of pulsating wire feeding.

Experiments on the 50% duty cycle and 40 Hz frequency of pulsating wire feeding speed control group were carried out. Figure 11 shows that for pulsed wire feeding, when there is no withdrawal, the transfer of droplets is rejection transition and short-circuit transition hardly occurs. After increasing the withdrawal, the proportion of rejection transition decreases significantly, and the proportion of short-circuit transition increases significantly. With the increase of withdrawal speed, the ratio of rejection transition increases significantly, and the proportion of short-circuit transition begins to decrease. Under the condition of 500 m/h withdrawal speed, the rejection ratio of pulsating wire feeding is the smallest and the short-circuit transition ratio is the largest. Whether or not there is withdrawal, the splash ratio hardly changes and the proportion is small. For pulsating wire feeding, the increase of wire withdrawal speed can reduce the short-circuit transition ratio and increase the rejection transition ratio.

As shown in Figure 12, the forming fraction of the 0.5 mm and 1.0 mm withdrawal quantities decreases with the increase of pulsating frequency. The general rule is that the forming fraction of the 0.5 mm withdrawal quantity group is higher than that of the 1.0 mm withdrawal quantity group at the same withdrawal speed. As shown in Figure 13, the coefficient of variation does not change with the rise of pulsating frequency, and the coefficient of variation of the 0.5 mm withdrawal quantity group is less than that of the 1.0 mm withdrawal quantity group.

Under the precondition of eliminating the influence of wire withdrawal speed, it can be seen in Figure 14 and Figure 15 That, for rejection transition, the transition ratio increases first and then decreases with the increase of pulsating wire feeding frequency at the 0.5 mm withdrawal quantity, and the overall trend is upward. At 40 Hz, the rejection transition ratio reaches a maximum of nearly 80%. At the 1.0 mm withdrawal quantity, the rejection transition ratio increases with the increase of pulsating wire feeding frequency. However, the absolute value is smaller than that of the 0.5 mm withdrawal, and the rejection transition ratio is more than 50% at 60 Hz pulsating frequency. The rejection transition ratio of pulsating wire feeding under the 0.5 mm withdrawal quantity is significantly higher than that under the 1.0 mm withdrawal quantity. For short-circuit transition, the transition ratio decreases with the increase of the pulsating frequency at 0.5 mm and 1.0 mm withdrawal quantities, but the slope of the 1.0 mm withdrawal is smaller than that of the 0.5 mm withdrawal. At the same frequency, the short-circuit transition ratio is significantly higher under the 1.0 mm withdrawal quantity than that under the 0.5 mm withdrawal quantity. The spatter ratio is negatively correlated with pulsating wire feeding frequency under the 0.5 mm and 1.0 mm withdrawals. From a numerical point of view, the spatter ratio under the 0.5 mm withdrawal is lower than 30% and the minimum value is 3.1% at 40 Hz. Meanwhile, the spatter ratio under the 1.0 mm withdrawal is higher than 27% and the maximum value is close to 35%. It can be seen that the welding spatter ratio of the 0.5 mm withdrawal quantity is significantly less than the 1.0 mm withdrawal quantity.

On the premise of excluding the influence of pulsating wire feeding frequency, the relationship between the droplet transfer ratio and the speed of pulsating wire feeding is not obvious under different withdrawal quantities, as is shown in Figure 16 and Figure 17. The rejection transition ratio under the 0.5 mm withdrawal quantity is larger than that under the same withdrawal speed; the short-circuit transition ratio under the 0.5 mm withdrawal quantity is smaller than that under the same withdrawal speed. The spatter generation rate has no clear trend.

The high-speed video and synchronous electrical signals show the droplet transition with pulsating withdrawal of 0.5 mm at 10 Hz and 50% duty cycle, as is shown in Figure 18. At 11,179 ms, the wire reburns to the highest point, the arc extinguishes at 11,197 ms, the droplet makes contact with the molten pool again at 11,219 ms, and the arc is ignited at that time.

Figure 19 shows the droplet transition process when the frequency is 10 Hz, the duty cycle is 50%, and the pulsating withdrawal is 1.0 mm. By observing the high-speed video camera and synchronous electrical signals, we found that when the frequency is 10 Hz, frequent arc extinguishing occurs at the 1.0 mm withdrawal quantity.

By studying the influence of pulsating wire feeding parameters on the droplet transfer process, we found that a mechanical force was produced during wire withdrawal, which promoted droplet transfer as well as the inertia force of droplets. In addition, the droplet rejection degree decreased and the droplet rejection transition ratio was inhibited when the wire was withdrawn, thus accelerating the droplet transfer process. Therefore, compared with pulsed wire feeding technology, pulsating wire feeding technology has great advantages in accelerating droplet transfer process by changing the wire feeding mode to wire withdrawal.

In order to determine more systematically and accurately the best test parameters to be used, we adopted appropriate optimization methods to improve the test design on the basis of the above parameters. The Taguchi methodology was utilized for planning the experiments and for modelling by using a specially designed experimental matrix known as the orthogonal array [27,28]. For this work, we will use Taguchi’s L9 orthogonal array to organize the experiments. Pulsating feeding frequency, withdrawal speed, and withdrawal quantity were selected as experimental parameters, and forming points, coefficient of variation, and transition ratio were considered as output responses. The optimal parameter range was then obtained. In the next step, we synthetically compared and analyzed optimization methods such as topology [29] or the iterative gradient search method [30,31] to determine the optimal parameters.

## 4. Conclusions

The effect of pulsating wire feeding parameters on droplet transfer in wet welding of flux-cored wire was studied, and the optical method based on X-ray transmission was used to explore this effect. The conclusions are as follows:
When welding at 10 Hz frequency, the phenomenon of reburning and arc breaking still occurs, but the welding forming effect is the best. At 60 Hz, the weld forming score is close to 0, the weld is extremely unstable, and it is difficult to produce continuous and stable welds. The short-circuit transition ratio decreases steadily with the increase of pulsating wire feeding frequency, the rejection transition ratio first rises and then decreases, and the splash ratio first decreases and then rises.When no withdrawal occurs, the transfer of droplets is dominated by rejection transition. With the increase of withdrawal, the proportion of rejection transition decreases and the proportion of short-circuit transition increases. With the increase of wire withdrawal speed, the proportion of short-circuit transition decreases and the proportion of rejection transition increases. However, the splash ratio hardly changes, and the proportion is small.The forming points of the 0.5 mm and 1.0 mm withdrawals decrease with the increase of pulsating frequency. The rejection transition ratio of pulsating wire feeding under the 0.5 mm withdrawal quantity is significantly higher than that under the 1.0 mm withdrawal quantity. The short-circuit transition ratio and spatter transition ratio are significantly higher under the 1.0 mm withdrawal quantity than those under the 0.5 mm withdrawal quantity.

## Figures and Tables

**Figure 1 materials-12-01715-f001:**
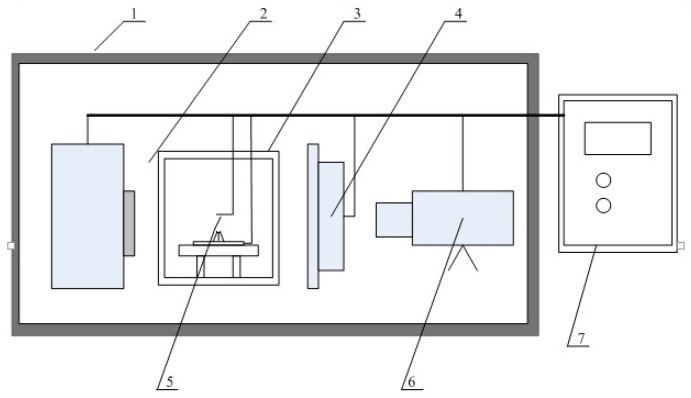
Composition of underwater welding X-ray high-speed camera system. 1—lead housing; 2—X-ray source; 3—test bench; 4—X-ray image intensifier; 5—welding wire; 6—high-speed camera; 7—console.

**Figure 2 materials-12-01715-f002:**
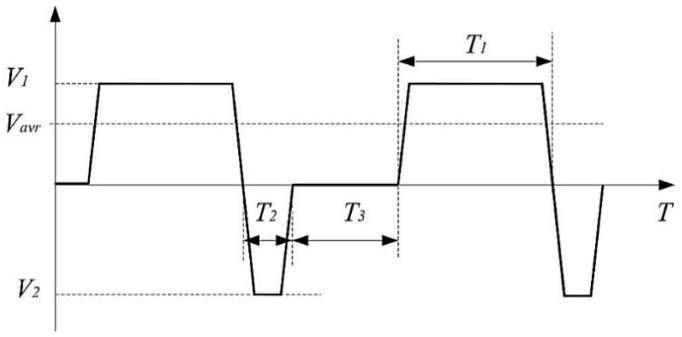
Wave form of wire feeding speed (feed withdrawal mode).

**Figure 3 materials-12-01715-f003:**
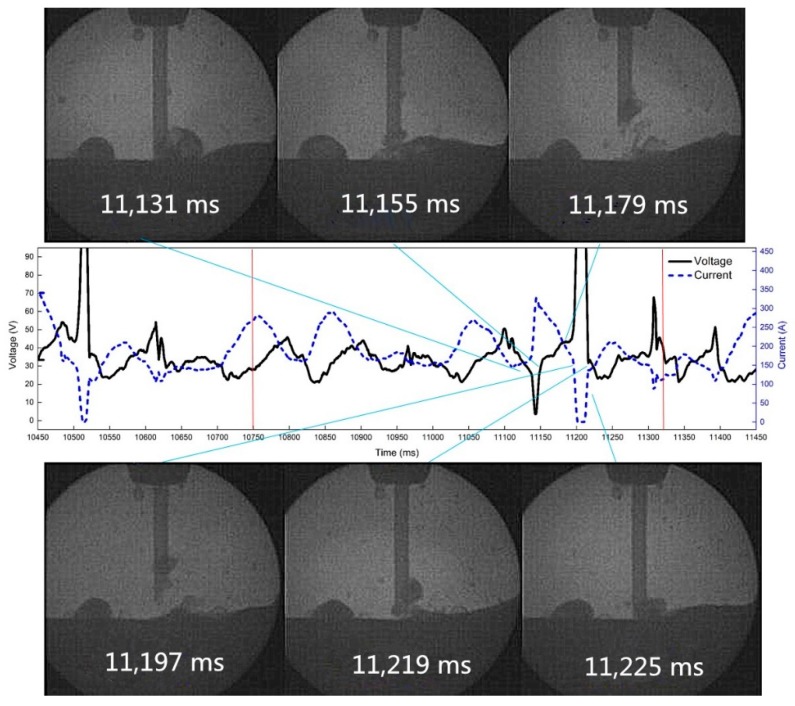
Droplet transition process at 10 Hz and 0.5 mm withdrawal quantity.

**Figure 4 materials-12-01715-f004:**
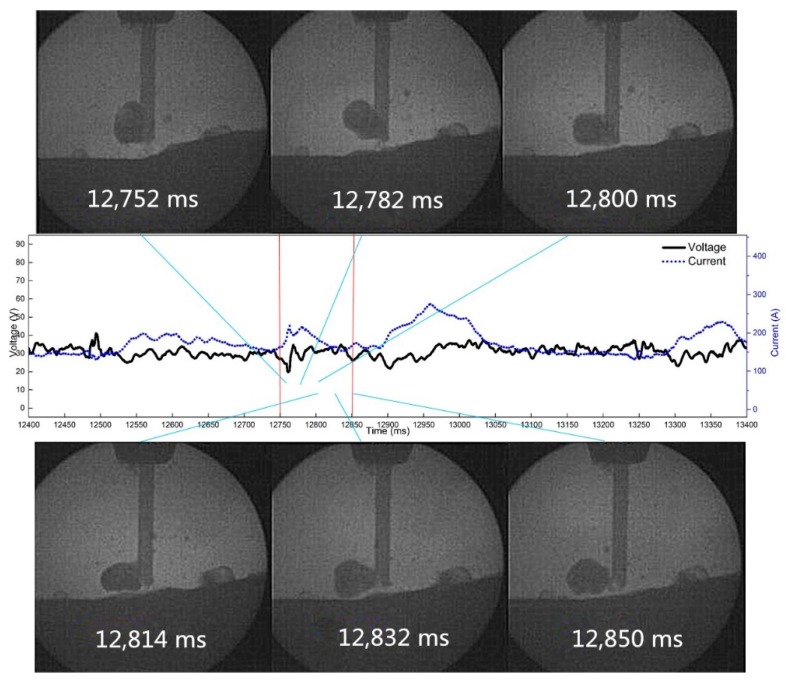
Droplet transition process at 40 Hz and 0.5 mm withdrawal quantity.

**Figure 5 materials-12-01715-f005:**
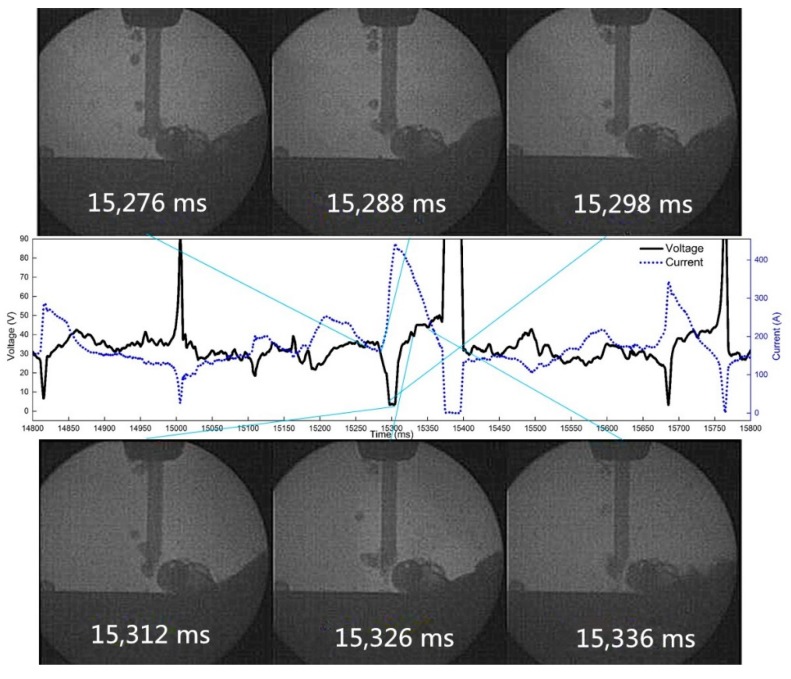
Droplet transition process at 60 Hz and 0.5 mm withdrawal quantity.

**Figure 6 materials-12-01715-f006:**
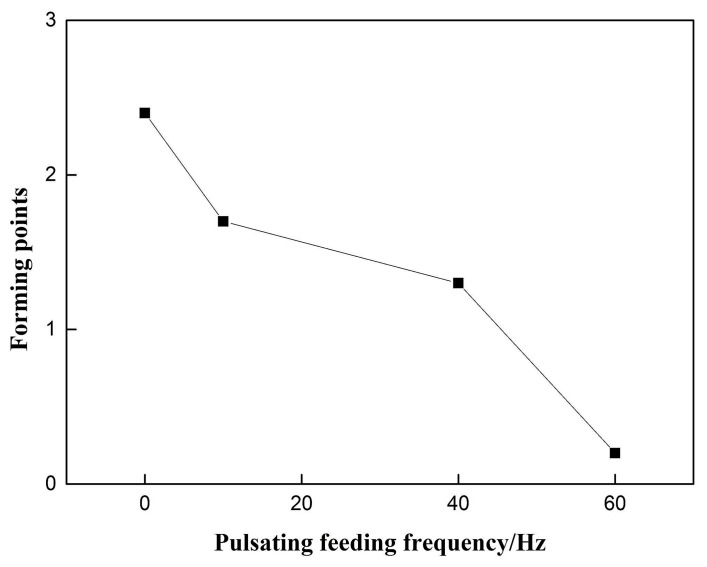
Effect of pulsating feeding frequency on weld forming.

**Figure 7 materials-12-01715-f007:**
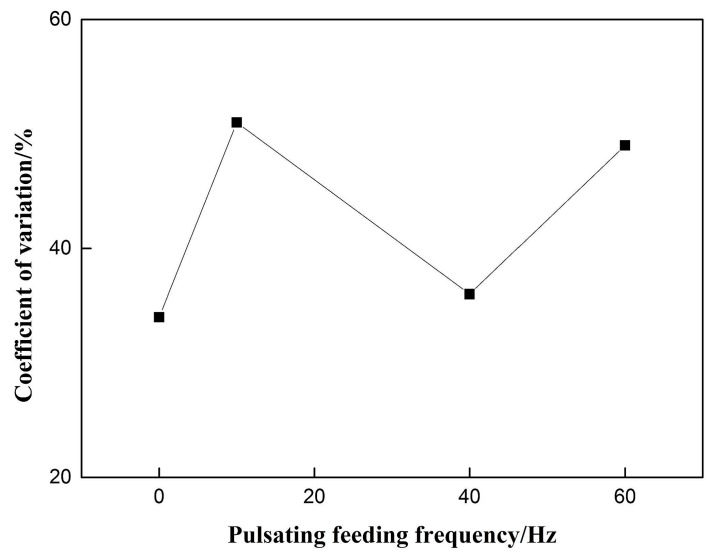
Effect of pulsating feeding frequency on coefficient of variation.

**Figure 8 materials-12-01715-f008:**
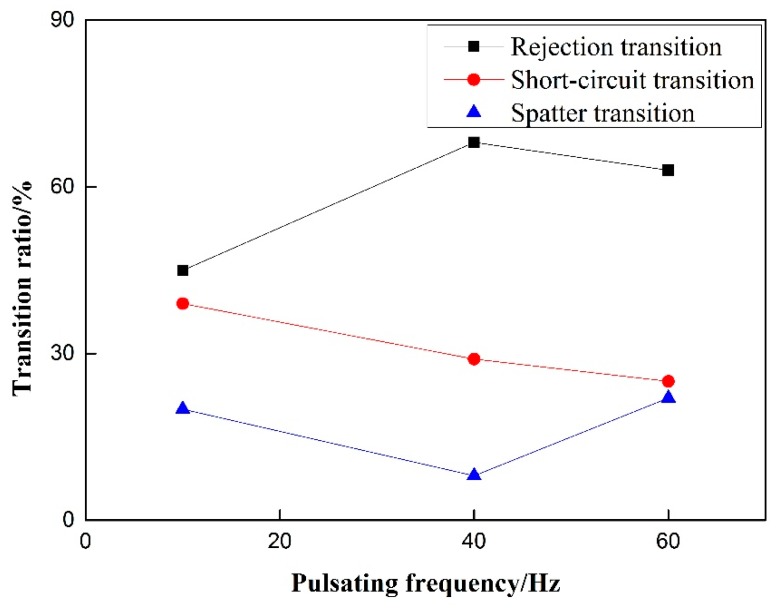
Effect of pulsating feeding frequency on transition ratio.

**Figure 9 materials-12-01715-f009:**
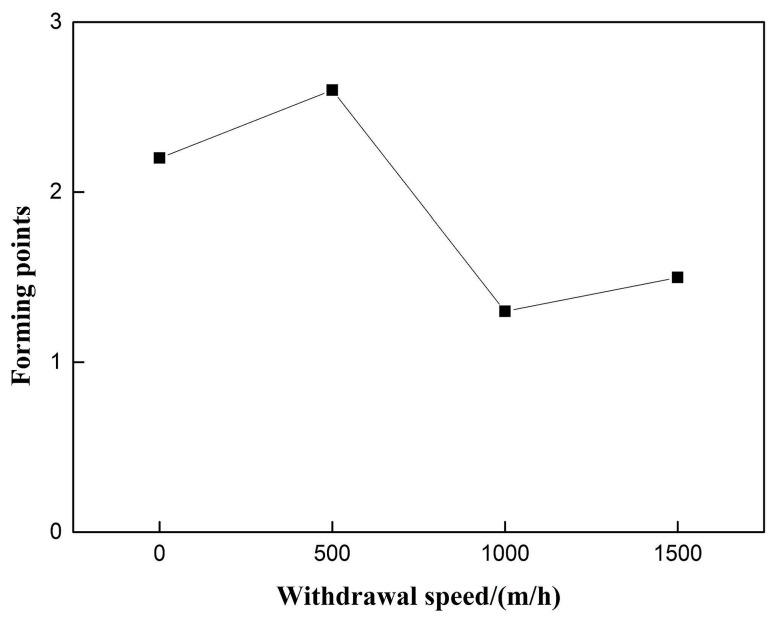
Effect of withdrawal speed on weld forming.

**Figure 10 materials-12-01715-f010:**
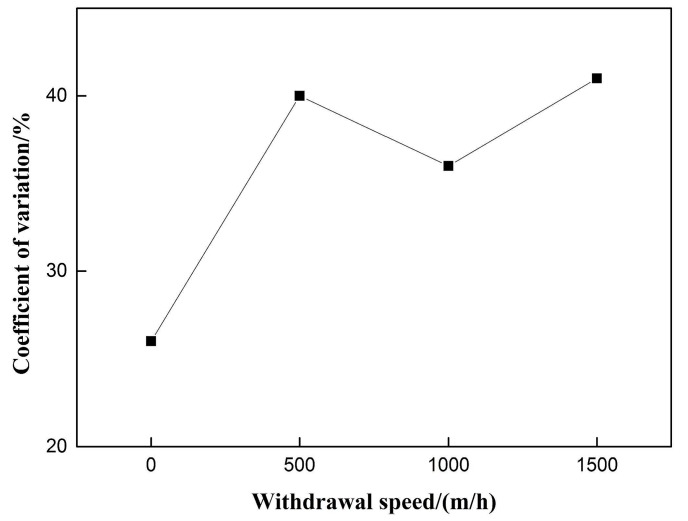
Effect of withdrawal speed on coefficient of variation.

**Figure 11 materials-12-01715-f011:**
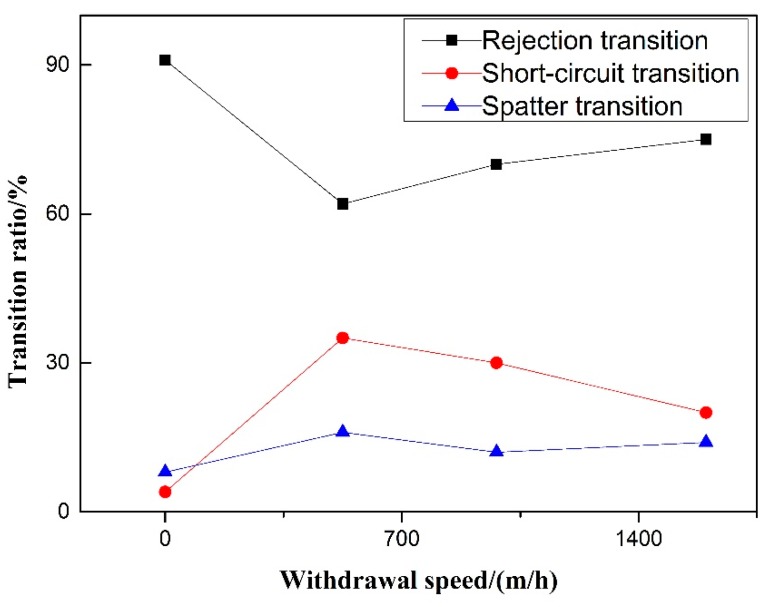
Effect of withdrawal speed on transition ratio.

**Figure 12 materials-12-01715-f012:**
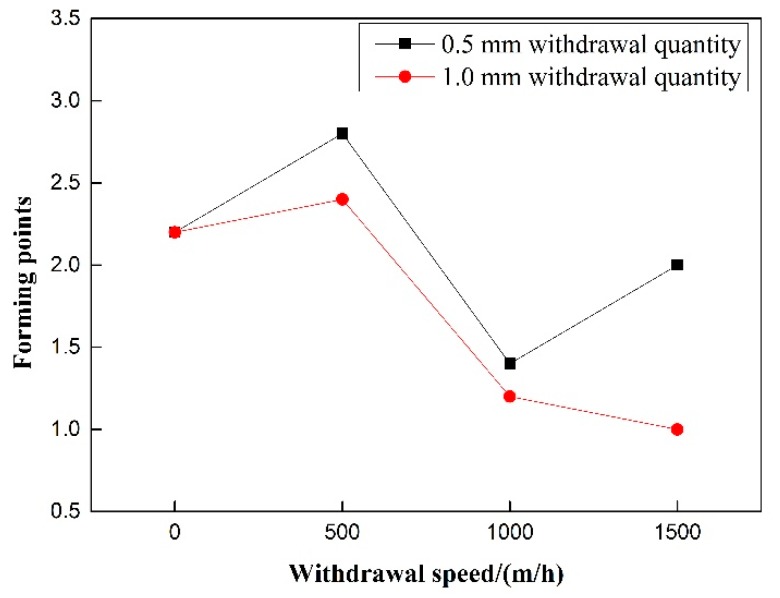
Effect of withdrawal speed on weld forming under different withdrawal quantity.

**Figure 13 materials-12-01715-f013:**
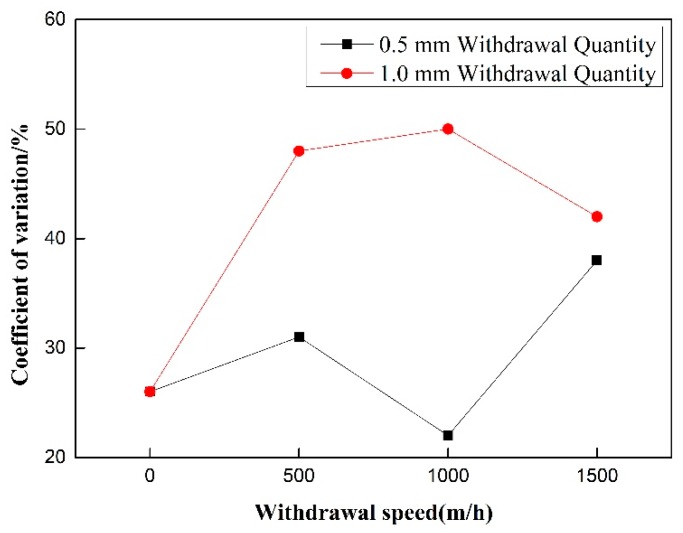
Effect of withdrawal speed on coefficient of variation under different withdrawal quantity.

**Figure 14 materials-12-01715-f014:**
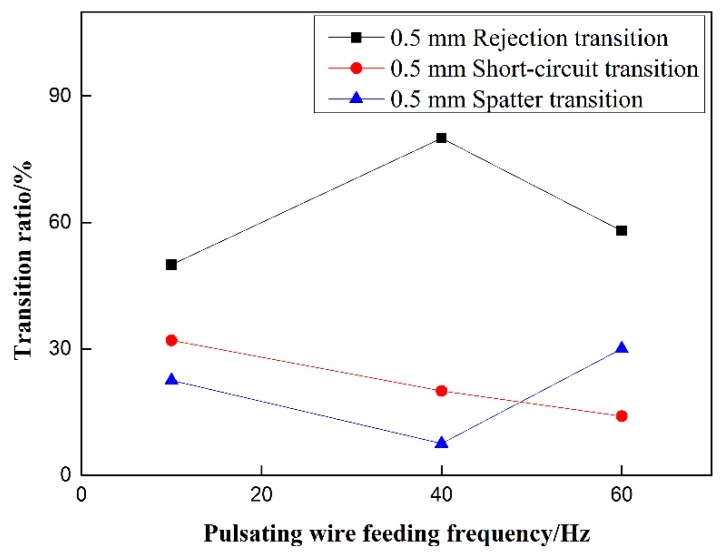
Effect of pulsating wire feeding frequency on transition ratio under 0.5 mm withdrawal quantity.

**Figure 15 materials-12-01715-f015:**
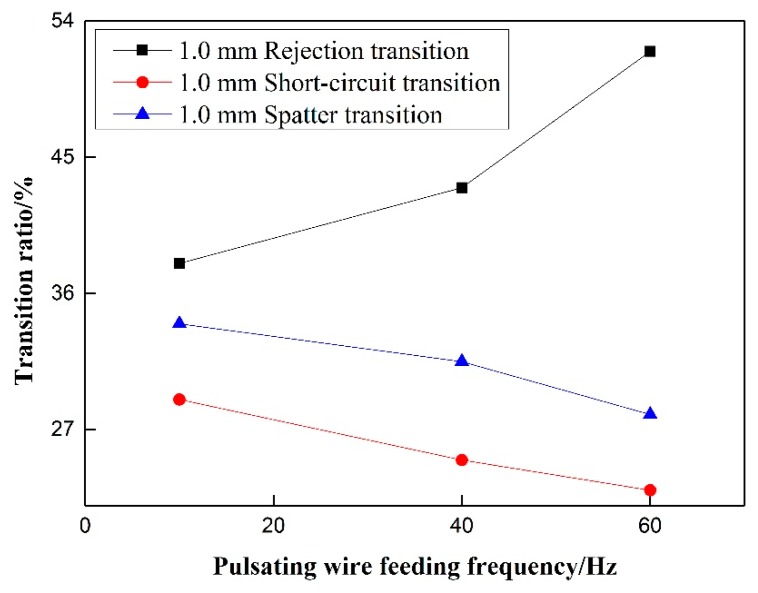
Effect of pulsating wire feeding frequency on transition ratio under 1.0 mm withdrawal quantity.

**Figure 16 materials-12-01715-f016:**
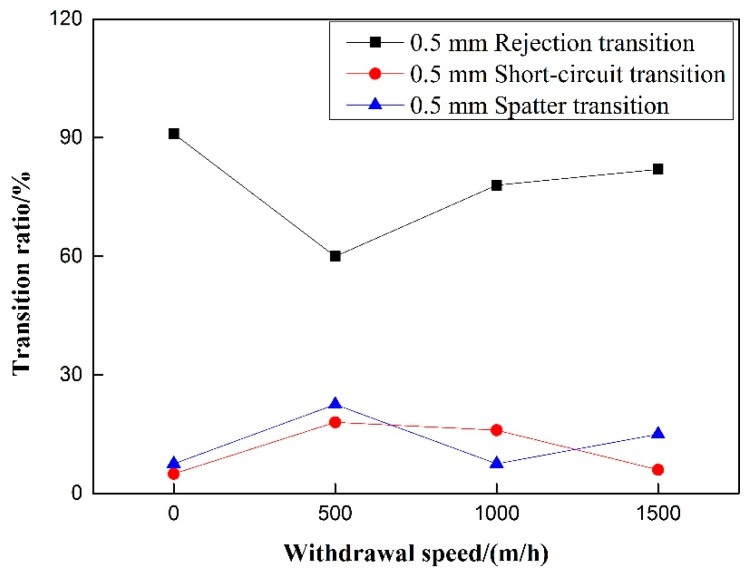
Effect of withdrawal speed on transition ratio under 0.5 mm withdrawal quantity.

**Figure 17 materials-12-01715-f017:**
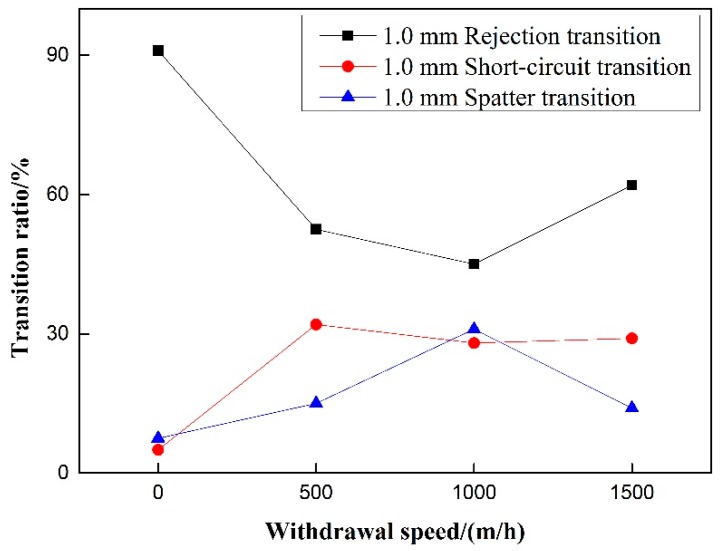
Effect of withdrawal speed on transition ratio under 1.0 mm withdrawal quantity.

**Figure 18 materials-12-01715-f018:**
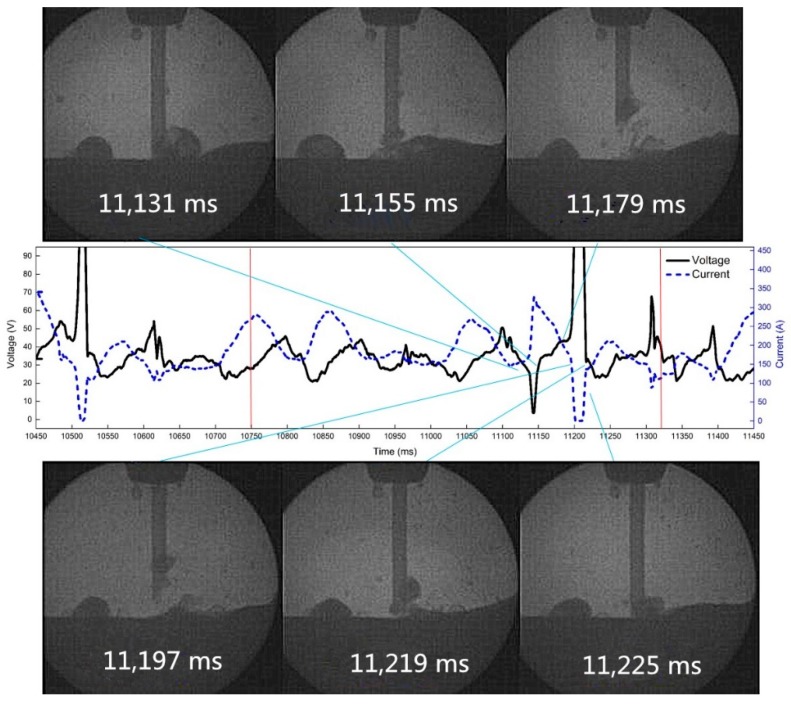
The arc extinguishing process of droplet reburning at 10 Hz and 0.5 mm withdrawal quantity.

**Figure 19 materials-12-01715-f019:**
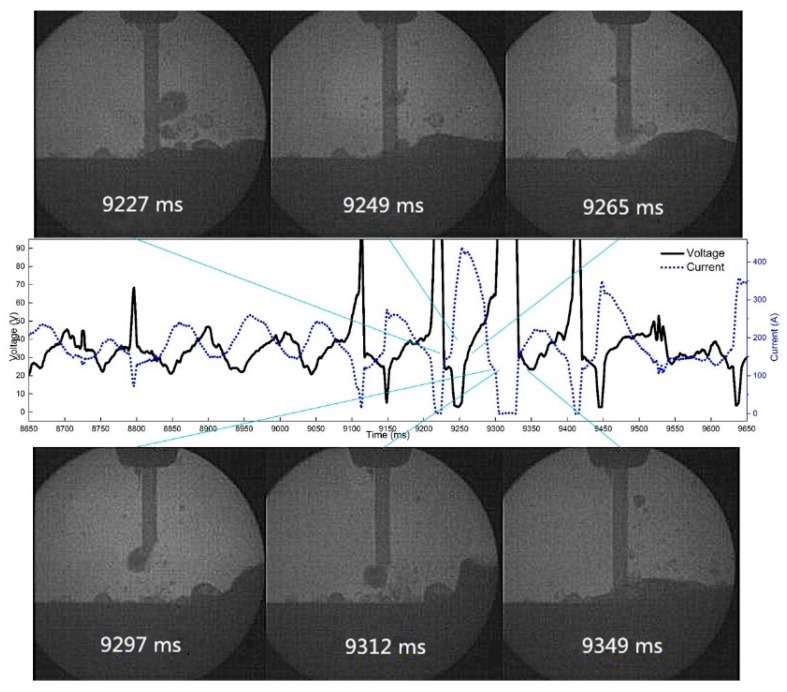
The arc extinguishing process of droplet reburning at 10 Hz and 1.0 mm withdrawal quantity.

**Table 1 materials-12-01715-t001:** Chemical composition of Q235 steel.

Material	C	Si	Mn	Cr	P	S	Ni	Fe
Q235	0.35	0.35	1.75	0.3	0.03	0.008	1.96	Remaining

**Table 2 materials-12-01715-t002:** Experimental group parameters of the pulsating wire feeding mode.

No.	Wire Feeding Speed/(m × h^−1^)	Frequency/Hz	Duty Cycle/%	Withdrawal Quantity/mm	Withdrawal Speed/(m × h^−1^)
1	200	10	50	0.5	1000
2	200	10	50	1.0	1000
3	200	40	50	0.5	1000
4	200	40	50	1.0	1000
5	200	60	50	0.5	1000
6	200	60	50	1.0	1000
7	250	10	50	0.5	1000
8	250	10	50	1.0	1000
9	250	40	50	0.5	1000
10	250	40	50	1.0	1000
11	250	60	50	0.5	1000
12	250	60	50	1.0	1000
13	150	40	50	0.5	1000
14	200	40	50	0.5	500
15	200	40	50	1.0	500
16	200	40	50	0.5	1500
17	200	40	50	1.0	1500

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
