# Peer review of "Control of Droplet Transition in Underwater Welding Using Pulsating Wire Feeding"

_materials, 2019, doi:10.3390/ma12101715_

Round 1

Reviewer 1 Report

Dear Authors,

Your article is a part of a global cycle of work related to the improvement of weldability of steel in the water environment. One of the trends is the analysis of the stability of welding arc under water. In my opinion, the work is interesting and valuable. However, it requires a several important improvements. I recommend: Major revision.

General remarks

I propose change the numbers of figures to. Fig. 1, Fig 2., etc. It will be more readable. Check correctness of figures numbering (there are two figures 3-10)

Move dots after brackets with citations.

Add spaces before units (in figures too).

Add references to formulas (if it is justified).

Use uniformly term: “pulsating withdrawal speed” or “withdrawal speed”.

Title: I propose to change to: Study on the control of droplet transition in underwater wet flux cored arc welding by pulsating wire feeding

Introduction

Literature review is well written, but some paragraphs require literature sources. For example, you provide information about the worsening of weldability of steel (metallurgy, diffusible hydrogen content, cold cracking) and without citation of recent references. Due to the aims&scope of the Materials journal, you can add a brief information about materials that are currently can be welded under water: mild steels, high strength steels, alloy (austenitic and duplex) steels and dissimilar joints (with relevant references).

Lines 36-38: and lines 40-43:

Please add references to each of this important information. Is there any possibility to improve underwater welding processes and avoid presented factors?

Line 61: I have not found in the literature of the article "Pokhonya" -line 61 (I guess it can be about Professor Pokhodnya from E. O. Paton Electric Welding Institute in Ukraine).

Line 98; Change Barton Welding Institute to E. O. Paton Electric Welding Institute.

Line 133: change 3-1 to 1.

Table 3: change 1 to 1.0 in 5th column.

Materials and Methods

Please show the chemical composition and mechanical properties of materials used (mainly: trade name of the wire).

Lines 99-100: Change “are” to have been” or “was”.

Effect of pulsating wire feeding on droplet transition

Check whether the chapter title should be: “Results” according to the magazine's guidelines.

You should add information about welding parameters as arc voltage and welding current, used in your experiments. These parameters have strong impact on the stability of welding arc.

Figure 3-5., Figure 3-6. Figure 3-7……. – please improve the quality of presented figures.

Fig. 3-10: change withdrawing to withdrawal.

References

Please check correctness of citations. They are made with errors in the text and in references list. It is difficult to determine whether all sources have been cited. References: 4-5; 7-8; 10-11 – join this reference respectively.

Format uniformly references.

In references are only two positions from last three years. You should add some of the latest references presented results of investigations of arc stability during underwater welding (for example: Wang and Sun papers).

In addition, there is a lack of works from Materials, and yet the journals from the MDPI publishing house are known for the fact that articles often form a cycle, which favors a scientific discussion. Searching “underwater welding”phrase in Title/Keyword shows that there are articles in this topic in the MDPi publishing house that have been latterly published.

Author Response

Dear reviewer, thank you for reviewing our manuscript in your busy schedule. According to your suggestion, we realized our shortcomings and made a lot of revisions to the manuscript. We also responded to your suggestion point by point. Please refer to the uploaded PDF.

Reviewer 2 Report

Please find the comments in the attachment.

Author Response

(The authors gave the same response as above.)

Round 2

Reviewer 1 Report

Dear Authors,

Thank you for considering most of my comments. However, in several places, either because of a misunderstanding or a short time to make changes, it did not happen according to my proposals. I recommend publishing the article after Major revisions, because I have some important remarks:

1.       Your article published in WiW (2017) in the title has "pulsing". Would not this term be appropriate now?

2.       Change “welding voltage” to “arc voltage" in the whole text

3.       Add spaces before units in figures (3-2 to 3-4; 3-11- to 3-16) – if possible

4.       Line 36: add “covered” before “electrode”

5.       Line 37: correct term is: “underwater wet self-shielded flux cored arc welding”

6.       Line 62; change “weld” to “make welded joint of”

7.       Line 69: check correctness of citations. There is no Prof Pokhodnya article in References.

8.        As is stated in the relevant literature, the biggest problem in underwater welding is high susceptibiity to cold cracking. I can't find this information in your article. Please add this important information (in line 45?).  You can also answer a question from Round#1 of revision - is there any possibility to avoid underwater problems? You can use this or a similar course of reasoning: The presence of a critical diffusible hydrogen amount in the steel welded joints causes the formation of cold cracks [ref,ref]. Methods for preventing cracking in the water environment are limited compared to in air welding. For this purpose, it can be used the temper bead technique [,,] and, to a limited extent, preheating [,] and control the diffusible hydrogen content by changing the values of welding parameters [,].

9.       Add information about welding polarity: DC+?

10.   In the "Detailed response to the Reviewer" you have written that you have changed the numbers of the figures, but in the manuscript there are still numbers as Fig 2.1, etc.

11.   The quality of used figures (listed in round #1 of my revision) is still poor. You should magnify size of numbers in Fig. 2-1.

12.   Line 112: You don’t need to use capital letters

13.   Lines 157, 171: add dots after (3-1) and (3-4).

14.   Line 119: change: “volts” to V

15.   Line 129: Does “arc breaking” mean: “arc extinguishing”?

16.   You cite only 6 references from last 5 years. 8 literature sources are extremely difficult to achieve: 9, 11, 16, 21, 23, 24, 25, 26.

17.   You should to add references from Materials journal. Searching “underwater welding” in Title/Keyword in Mdpi website shows that there are articles in the Materials that have been latterly published and showed underwater weldability issues.

18.   Remove form references letters from brackets; [J], [M],[D] and format references according to journals guidelines.

Best regards

Author Response

Dear Editors and Reviewers: 

Thank you very much for giving us an opportunity to revise our manuscript, we appreciate editors and reviewers very much for your positive and constructive comments and suggestions on our manuscript.

We have studied reviewer’s comments carefully and have made revision which marked in yellow in the paper. We have tried our best to revise our manuscript according to the comments. Attached please find the revised version, which we would like to submit for your kind consideration.

We would like to express our great appreciation to you and reviewers for comments on our paper. Looking forward to hearing from you.

Thank you and best regards.

Reviewer 2 Report

The authors have addressed most of the raise issues. Improvements can be seen in the revision. However, some important comments on the original manuscript have not been fully addressed. The authors need to respond to the reviewer and readers by discussing the arising issues in both response letter and manuscript. The review suggests a minor revision before publication.

1)      The authors should adopt the optimization methods in determining the optimal parameters or discuss those methods explicitly as future works in the manuscript with proper references, for benefits and convenience of readers.

Please refer to the following references for more information on Taguchi method, iterative gradient search, exhaustive search, topology optimization, etc.

https://doi.org/10.1177/0954405415572662

https://doi.org/10.1007/s00170-018-2508-6

https://doi.org/10.1016/j.eswa.2014.03.037

https://doi.org/10.1007/s00158-016-1565-4

In addition, in response, the orthogonal method is not an optimization method. Please study the above references.

2)      Figures 3-3, 3-4, 3-17, 3-18 are not readable (the middle sections). What are the horizontal axis, vertical axis, and legends respectively?

3)      Please check the format of references, and ensure they comply with the standard of Materials Journal. Journal name abbreviation, author name abbreviation, etc.

Author 1, A.B.; Author 2, C.D. Title of the article. Abbreviated Journal Name YearVolume, page range. Available online: URL (accessed on Day Month Year).

Author Response

(The authors gave the same response as above.)

Round 3

Reviewer 1 Report

Dear Authors,

Thank you very much for introducing my remarks to the text. The article is currently being read very well, however, I have found some technical errors, so I recommend Minor revision.

I would also like to point out that citing hard-to-access works, internal reports and translation of titles from national languages [for example: 11, 18, 29] makes it difficult for the reader to reach information, and databases (WoS, Scopus, GS) can not correctly save information. Of course, in some cases this is justified and necessary due to the pioneering nature of these works. However, in the case of the description of obvious technological processes and the provision of generic information such as: pre-heating, hydrogenation of deposited metal, cold cracking, underwater welding metallurgy etc. much better literature sources can be found.

In addition, some of the sources are not accurately cited. For example, in the article Prof. Skorupa from Poland [9] the topic of hydrogen is only discussed marginally. The article by Aloraier et al. [15] does not apply to underwater welding, so I suggest you remove the fragment (lines 65-67).

Similarly, in line 75, articles 19 and 20 were cited, which deal with welding in the air, not under water, as the whole fragment suggests. Due to their difficult accessibility, I can not check the correctness of citing works: 30-32.

For this reason, I can suggest changing some sources. Readily available, published in JCR journals or open acces and matching the subject matter are, for example, articles:

Zhang, H. T., Dai, X. Y., Feng, J. C., & Hu, L. L. (2015). Preliminary investigation on real-time induction heating-assisted underwater wet welding. Welding Journal1, 8-15.

Kong, X., Li, C., Zou, Y., Zhang, J., Hu, Y., & Wang, J. (2016). Measurement and analysis of the diffusible hydrogen in underwater wet welding joint. In MATEC Web of Conferences(Vol. 39, p. 03004). EDP Sciences.

Świerczyńska, A., Fydrych, D., & Rogalski, G. (2017). Diffusible hydrogen management in underwater wet self-shielded flux cored arc welding. International Journal of Hydrogen Energy42(38), 24532-24540.

Fydrych, D., Świerczyńska, A., Rogalski, G., & Łabanowski, J. (2016). Temper bead welding of S420G2+ M steel in water environment. Advances in Materials Science16(4), 5-16.

Gao, W., Wang, D., Cheng, F., Di, X., Deng, C., & Xu, W. (2016). Microstructural and mechanical performance of underwater wet welded S355 steel. Journal of Materials Processing Technology238, 333-340.

Maksimov, S. Y. (2010). Underwater arc welding of higher strength low-alloy steels. Welding International24(6), 449-454.

Santos, V. R., Monteiro, M. J., Rizzo, F. C., Bracarense, A. Q., Pessoa, E. C. P., Marinho, R. R., & Vieira, L. A. (2012). Development of an oxyrutile electrode for wet welding. Welding Journal91(12), 319-328.

I have a few technical, editorial remarks:

Line 45: should be: “diffusible hydrogen content in deposited metal” or “diffusible hydrogen content in weld metal”

Line 47: should be “cooling rate”

Line 55:should be: “components in covering”

Line 61: should be “Tomków et al.”

References: check [8] (Design), [10] (Welding and Joining), [17] (Journal), [18] (Shi nameless), [29] (Machine), [31] (Technology).

Author Response

Dear Reviewers:

Thank you very much.According to your comment,we have had the manuscript polished and corrected the mistakes.

Best wishes.
